# Phylogenetic Analysis of *Escherichia coli* Isolated from Australian Feedlot Cattle in Comparison to Pig Faecal and Poultry/Human Extraintestinal Isolates

**DOI:** 10.3390/antibiotics12050895

**Published:** 2023-05-11

**Authors:** Yohannes E. Messele, Darren J. Trott, Mauida F. Hasoon, Tania Veltman, Joe P. McMeniman, Stephen P. Kidd, Steven P. Djordjevic, Kiro R. Petrovski, Wai Y. Low

**Affiliations:** 1The Davies Livestock Research Centre, The University of Adelaide, Adelaide, SA 5371, Australia; yohannes.messele@adelaide.edu.au (Y.E.M.);; 2The Australian Centre for Antimicrobial Resistance Ecology, The University of Adelaide, Adelaide, SA 5005, Australia; 3Meat & Livestock Australia, Level 1, 40 Mount Street, North Sydney, NSW 2060, Australia; 4Research Centre for Infectious Disease, School of Biological Sciences, University of Adelaide, Adelaide, SA 5005, Australia; 5Australian Institute for Microbiology & Infection, University of Technology Sydney, Ultimo, NSW 2007, Australia

**Keywords:** *Escherichia coli*, genotype, epidemiology, phylogeny

## Abstract

The similarity of commensal *Escherichia coli* isolated from healthy cattle to antimicrobial-resistant bacteria causing extraintestinal infections in humans is not fully understood. In this study, we used a bioinformatics approach based on whole genome sequencing data to determine the genetic characteristics and phylogenetic relationships among faecal *Escherichia coli* isolates from beef cattle (n = 37) from a single feedlot in comparison to previously analysed pig faecal (n = 45), poultry extraintestinal (n = 19), and human extraintestinal *E. coli* isolates (n = 40) from three previous Australian studies. Most beef cattle and pig isolates belonged to *E. coli* phylogroups A and B1, whereas most avian and human isolates belonged to B2 and D, although a single human extraintestinal isolate belonged to phylogenetic group A and sequence type (ST) 10. The most common *E. coli* sequence types (STs) included ST10 for beef cattle, ST361 for pig, ST117 for poultry, and ST73 for human isolates. Extended-spectrum and AmpC β-lactamase genes were identified in seven out of thirty-seven (18.9%) beef cattle isolates. The most common plasmid replicons identified were IncFIB (AP001918), followed by IncFII, Col156, and IncX1. The results confirm that feedlot cattle isolates examined in this study represent a reduced risk to human and environmental health with regard to being a source of antimicrobial-resistant *E. coli* of clinical importance.

## 1. Introduction

*Escherichia coli* is a Gram-negative, facultative anaerobic, rod-shaped bacterium commonly isolated as a normal component of the autochthonous microbiota of the animal and human gut [1]. However, many different *E. coli* pathotypes containing a diverse array of virulence genes (VGs) also exist, and these are capable of causing both intestinal and extraintestinal infections in both animals and humans [2]. Although many *E. coli* sub-types are relatively host-specific [3,4], some lineages have a broad host range and can potentially spread between humans, animals and their products, and the environment [5,6,7]. The ability of *E. coli* to spread to various hosts and environments whilst causing different infection types (i.e., both intestinal and extraintestinal infections) emphasizes the importance of monitoring spread and taking measures to regulate it. The most common methods to regulate the spread of specific *E. coli* pathotypes include the judicious use of antimicrobial agents, following good hygiene practices, and limiting the movement of infected animals and/or their contaminated products through strict biosecurity [8].

Antimicrobials have played a crucial role in the treatment and management of both intestinal and extraintestinal *E. coli* infection in animals and humans. However, selective pressure exerted by antimicrobials is assumed to be one of the main factors in the emergence and spread of antimicrobial resistance (AMR) among both pathogenic and commensal *E. coli* [9]. Although there is a clear link between the use of antimicrobials and the development of resistance among *E. coli*, there are also other factors in play such as the transmission rate of pathogenic subtypes to different hosts, the rates at which pathogenic lineages mutate and/or exchange accessory genes with commensals, and the selection of co-resistance to unrelated antimicrobials [10,11]. 

Both antimicrobial resistance genes (ARGs) and VGs may be easily transferred between *E. coli* strains because they are often located within transferable genetic elements such as bacteriophages, genomic islands, insertion sequences (ISs), integrons, transposons, and plasmids [12]. *E. coli* can also easily acquire mobile genetic elements containing these genes from other closely related bacteria. Pan genome analyses of *E. coli* isolates from multiple hosts have shown the ability of this species to rapidly evolve through gene acquisition and genetic modification whilst largely retaining a clonal population structure [5]. Additionally, a genetic similarity between certain broad host range bacterial clonal lineages of *E. coli* (such as ST10, ST58 and ST648) isolated from humans and animals suggests that frequent cross-species transmission is probable for these subtypes [13,14,15].

Whilst many studies have identified virulence and ARGs in *E. coli* isolated from various animal and human populations [16,17,18,19,20,21,22,23], few studies have undertaken valid comparisons between these diverse groups. Hence, little is known to what extent different livestock-associated commensal *E. coli* seed the environment and contribute to the AMR burden relevant to human health. Therefore, regular surveys are crucial for the effective management of AMR and infection control for both animal and human health. Australia has strict registration and regulation protocols for antimicrobial use in livestock production systems [24,25], and represents a unique study site given that there are also bans in place on the importation of live, food-producing animals, strict quarantine laws, and no land borders with other countries. Although Australia has implemented measures to prevent AMR, the spread of AMR is a complex issue that requires ongoing monitoring throughout the food chain. Regular monitoring and surveillance of animal production systems aims to identify AMR hot spots that may need immediate attention to implement effective prevention strategies [26,27]. 

The monitoring of AMR in *E. coli* can take place at various levels, including at the farm, environment, retail, and consumer level. In our previous study, tetracycline resistance increased by 17.8% in faecal-origin *E. coli* isolated from healthy cattle at feedlot entry compared to exit [28]. The prevalence of extended-spectrum β-lactamase (ESBL)-producing *E. coli* also rose from 0.7% to 4.0% between entry and exit samples. However, any possible links between these isolates and the *E. coli* isolated from other food animals and humans in Australia has not yet been determined. Therefore, this study compared the genetic similarity, ARG, plasmid, and VG profile of these cattle-origin *E. coli* isolates to contemporaneous isolates from other food animals and humans in Australia. This was undertaken to assess both the risk of animal-to-human isolate transfer via direct contact, food, and/or the environment, and the potential for horizontal gene transmission and spread among the various hosts.

## 2. Results

### 2.1. Phylogroup, Sequence Types and SNP Analyses

The *E. coli* isolates were distributed into phylogroups predominantly according to their host species of origin, with most cattle and pig isolates belonging to A and B1 and most poultry and human isolates belonging to B2 and D (Table 1). 

Further genomic analyses revealed a high diversity of sequence types (STs) among the beef cattle *E. coli* isolates (18 known STs), the most frequently identified being broad host range ST10 (27.0%), ST278 (18.9%), and ST109, ST515, ST641, and ST2035 (5.4% each). The remaining 12 cattle STs were represented by single isolates only. The most common STs identified among the pig (faecal), poultry (extraintestinal), and human (extraintestinal) *E. coli* isolates were ST361, ST117, and ST73, respectively. As expected for extraintestinal isolates, most poultry and human isolates belonged to STs located within phylogroups B2 and D. (Figure 1; Appendix A). By comparison, most cattle and pig faecal isolates belonged to STs associated with commensal phylogroups A and B1. In this way, most cattle and pig isolates were distributed into two clades (Clade 1 containing phylogroup B1 isolates, including those belonging to ST278, and Clade 2 containing phylogroup A isolates, including those belonging to ST10). Clade 2 ST10 isolates were more genetically diverse compared to Clade 1 ST278 isolates. Additionally, occasional human or poultry extraintestinal clinical isolates were found in Clade 1 (three isolates, one human and two poultry, all representing single STs) or Clade 2 (a single human ST10 isolate that was genetically distinct from cattle isolates).

### 2.2. Antimicrobial Resistance Genes

Overall, 60 ARGs were identified in the 141 *E. coli* isolates, encoding resistance to aminoglycosides (fourteen ARGs), amphenicols (four ARGs), ß-lactams (twenty-one ARGs), fluoroquinolones (three ARGs), folate synthesis inhibitors (ten ARGs), macrolides (five ARGs), and tetracyclines (three ARGs). The mean number of ARGs present in isolates obtained from each host species were: beef cattle (n = 3), poultry and human (n = 4), and pig (n = 7). Most pig *E. coli* isolates were MDR (extremely high: 42/45; 93.3%) compared to humans (very high: 24/40; 60.0%), poultry (high: 9/19; 47.4%), and beef cattle isolates (high: 14/37; 37.8%) (Figure 2). The study investigated pig and poultry isolates known to have the Class 1 integrase gene *incl1*, but the gene was also found in 10.8% (4/37) of the beef cattle isolates and 67.5% (27/40) of the human extraintestinal isolates. Among the *intI1*-containing *E. coli* isolated from beef cattle, three were originally isolated from ESBL selective agar. The study found that beef cattle *E. coli* isolates carrying the *intI1* gene were more likely to contain multiple ARGs, including those encoding resistance to extended-spectrum ß-lactams, and were identified as ST88, ST1102, ST9967, and ST540 (Appendix A).

#### β-Lactam Resistance Genes

A high proportion of the total isolate collection possessed the aminopenicillin resistance gene *bla*_TEM-1B_ (70/141; 49.6%), with the next most commonly identified genes being *bla*_CTX-M-15_ (low prevalence: 6/141; 4.2%) and *bla*_CTX-M-27_ (low prevalence: 5/141; 3.5%), both imparting resistance to expanded-spectrum cephalosporins (Table 2). These were also the most prevalent β-lactam resistance genes identified among the beef cattle isolates (bla_TEM-1B_ [8/37, high prevalence: 21.6%], *bla*_CTX-M-15_ and *bla*_CTX-M-27_ [low prevalence: 3/37; 8.1% each]). The *bla*_CMY-2_ plasmid-mediated *AmpC* β-lactamase gene was also identified in two bovine isolates.

### 2.3. Disinfectant Resistance Genes

Only Quaternary ammonium compound (*qacE*) and hydrogen peroxide (*sitABCD*) resistance genes were detected among the 141 *E. coli* isolates (Figure 3). Both disinfectants are commonly used in hospitals, healthcare facilities, and the food industry. The highest proportions of disinfectant resistance genes were detected among poultry clinical isolates (*qacE* [12/19, very high: 63.1%] and *sitABCD* [10/19, very high: 52.6%], respectively) and human clinical isolates (*qacE* [18/40, high: 45%]; *sitABCD* [28/40, very high: 70.0%]). By comparison, the prevalence of disinfectant resistance genes was much lower among pig commensal isolates (both 9/45; moderate: 20.0%) and beef cattle commensal isolates (*qacE* [1/37, low: 2.7%] and *sitABCD* [2/37, low: 5.4%]).

### 2.4. Identification of Plasmid Replicons 

Overall, the PlasmidFinder analysis identified 35 different plasmid replicons in the 141 studied isolates (Figure 4). Twenty-seven Incompatibility (Inc) plasmid types were identified, the most common being IncFIB (AP001918) (89/141; 63.1%), IncFII (52/141; 36.9%) and IncX1 (25/141; 17.7%). Overall, 114/141 (80.8%) of the isolates contained more than two different plasmid replicons, with a high proportion of isolates (46/141; 32.6%) harbouring both IncFIB (AP001918) and IncFII replicons, whilst 5/141 isolates (3.5%) harboured IncFIB (AP001918), IncFII, and IncX1 replicons. Isolates that harboured two or more plasmid replicons included 24/37 beef cattle (64.8%), 37/45 pig (82%), 19/19 poultry (100%), and 34/40 human (85%) isolates. Seven Col plasmids containing genes responsible for the production of bacteriocins or colicins were also found in the isolate collection. These included Col156 (27/141; 19%), Col (BS512) (10/141; 7.1%), and ColpVC (6/141; 4.2%). Only 13/141 isolates (9.2%) did not possess any of the tested Inc replicons or colicin-associated genes. A total of 18 different plasmid types were identified among the beef cattle isolates, of which the most common were IncFIB (AP001918) (23/37; 62.2%), IncFII (16/37; 43.2%), IncFIC (FII) (8/37; 21.6%), and IncFIA, IncFII (pHN7A8), and IncI1-I (Alpha) (7/37; 18.9%, each). Among the IncF plasmids found in beef cattle, a total of 14 distinct pMLSTs were detected. A total of 8/37 beef cattle isolates (21.6%) did not contain any plasmids. The most frequently occurring pMLST was F89:A-:B- (5/14; 35.7%), followed by F89:A-:B-, F95:A-:B-, and F104:A-:B16 (3/14, 21.4% each). Among human isolates, the most commonly observed pMLST were F29:A-:B10 (8/40; 20%), F-:A-:B- (5/40; 12.5%), and F51:A-:B10 (4/40; 10%) (Appendix A).

### 2.5. Agreement between Plasmid Replicons and ARG Arrays

The most common plasmid replicons, IncFIB (AP001918) and IncFII, were co-associated with high proportions of multiple ARGs representing several classes, indicating the likelihood that these genes are co-located on specific dominant plasmids. These included the aminoglycoside ARGs *aph(3'')-Ib* and *aph(6)-Id*, ß-lactam ARG *bla*_TEM-1B_, tetracycline ARGs *tet(A)* and *tet(B)*, and folate synthesis inhibitor ARGs *dfrA5*, *sul2* and *sul1* (Figure 5).

### 2.6. Virulence Genes

A total of 103 VGs were identified in the 141 isolates; each isolate carried between 1 and 37 VGs (Figure 6). Eighty isolates (56.7%) had ≥10 VGs, which were distributed by phylogenetic group as follows: phylogroup B2 (29/141; 36.2%), B1 (13/141; 16.2%), D (12/141, 15.0%), A (11/141; 13.7%), E (6/141; 7.5%), G (5/141; 6.2%), F (3/141; 3.7%), and C (1/141; 1.2%). Isolates with the least VGs were mostly obtained from cattle and pigs and distributed into phylogenetic groups A and B1, whereas isolates with the most VGs associated with extraintestinal infection (as expected) were distributed into phylogenetic groups B2 and D, predominantly sourced from humans and poultry.

## 3. Discussion

Whilst some international studies have suggested that food-producing animals can be an important reservoir for virulent pathogens and their ARGs [29,30,31], other studies are more equivocal in their findings [32,33]. Given Australia’s unique animal production systems and their potential impact on the evolution of animal microbes in isolation from the rest of the world, the aim of this study was to compare the antimicrobial/disinfectant resistance, plasmid, and VG repertoires of commensal *E. coli* isolated from healthy beef cattle (n = 37) from a single feedlot, in comparison to previously characterised pig faecal (n = 45), poultry (n = 19), and human extraintestinal isolates (n = 40). This study had three major findings. First, the phylogenic analyses revealed that most beef cattle isolates clustered together with pig isolates in two main clades of related STs (within phylogroups A and B1), while most poultry and human isolates clustered together in phylogroups B2 and D. Second, while a significant proportion of beef cattle isolates carried plasmid-encoded extended-spectrum cephalosporin resistance genes (7/37; 18.9%), further analysis in comparison to human extraintestinal isolates suggested that they are of limited impact to human health. Third, a significant number of the beef cattle *E. coli* isolates (10/37; 27%) belonged to broad-host range ST10, including three isolates (30%) carrying genes encoding resistance to extended-spectrum cephalosporins.

The phylogenetic tree derived from the SNPs showed that isolates from beef cattle and pigs clustered together in phylogenetic groups (A and B1) not typically associated with highly virulent extraintestinal infection, whereas most poultry and human isolates were intermixed in separate clades in phylogroups B2 and D. This is consistent with previous research on poultry and human *E. coli* isolates, which suggested that clinical isolates from these sources share a high degree of genetic and pathotypic similarity, suggesting frequent anthropozoonotic exchange [34,35,36]. Additionally, studies conducted in Europe also confirmed that phylogroups A and B1 were most commonly represented by isolates obtained from healthy chicken, cattle and pig faeces [37,38]. In the present study, a single human extraintestinal isolate was identified as ST10; however, it was distinct from beef cattle and pig isolates within this genetically diverse, broad host range clonal lineage, which can be of clinical significance to human health [39].

The spread of pathogens that are resistant to third-generation cephalosporins is a critical concern that has extended beyond hospital environments. In this study, the plasmid-encoded resistance gene *bla*_TEM-1B_, which confers resistance to ß-lactam antimicrobials such as amoxicillin, was detected in 8/37 (21.6%) beef cattle isolates. Furthermore, the study found plasmid-mediated *bla*_CTX-M_ expanded-spectrum cephalosporin ARGs in 6/37 beef cattle isolates (16.2%) and a *bla*_CMY-2_ AmpC ß-lactamase gene in an additional isolate. Contrary to this, a study conducted in South Africa, which analysed isolates obtained from cattle faeces and raw beef samples, found a significantly higher prevalence of β-lactam resistance genes, including *bla*_TEM_ (85.5%), *bla*_SHV_ (69.6%), and *bla*_CTX-M_ (58.0%) [40]. In the present study, the amplification of beef cattle-origin isolates carrying ESBL or AmpC ß-lactamase ARGs in exit (4%) compared to entry (0.7%) samples may indicate that a proportion of the beef cattle population acquire these ARGs during the feeding period, which may be related to the reserve use of ceftiofur for bovine respiratory disease cases not responding to first or second line therapy, or those occurring late in the feeding period [25]. A previous study conducted in the United States found that administering ceftiofur to feedlot cattle can result in a selective pressure that increases the level of *bla*_CMY-2_ carriage [41]. 

There is widespread evidence from various studies conducted globally documenting the infrequent isolation of ESBL-producing *E. coli* in food animal populations such as beef cattle, pigs, and poultry [42,43,44]. However, there is currently no concrete evidence documenting the direct transmission of these bacteria from animals to humans via the food chain [45]. However, previous studies conducted in pig and poultry industries have reported a high level of ARG similarity between farmer-derived and animal-origin isolates, indicating close and direct contact as an important transmission vector [46,47,48,49,50,51,52]. Although the present study found *bla*_CTX-M-15_ in 3/40 (7.5%) human *E. coli* isolates, these appeared to be unrelated to the contemporaneous animal-origin ESBL genes, especially given the widespread use of ß-lactam antimicrobials in human medicine and the resulting selection pressure for the evolution of antimicrobial-resistant bacteria. 

In this study, 29.7% of beef cattle isolates were MDR, compared to 93.3% for pig, 47.4% for poultry, and 60.0% for human isolates. Even though the pig isolates were selected based on carriage of *intl1,* the higher proportion of multidrug resistance observed may be due to the increased use of prophylactic antimicrobial agents in pig farms and the management practices that facilitate the maintenance and dissemination of antimicrobial resistance determinants in both the hosts and the farm environment compared to poultry and cattle [53]. In this study, poultry isolates containing the *intl1* gene were also selected. Unlike the surveillance-derived pig isolates, these pathogenic poultry isolates showed a lower level of multidrug resistance. This discrepancy might be attributed to the possibility that pathogenic isolates could shed resistance genes in favour of virulence genes [54], the clonal nature of some isolates in different hosts [55], and variations in management practices and sample sizes [56]. By comparison, relatively low rates of multidrug resistance were observed among the cattle isolates given that the prophylactic administration of antimicrobial agents for bovine respiratory disease to large numbers of animals is rarely practiced in Australia and did not occur during this study at the host feedlot [25]. 

The majority of ARGs detected in the present study were likely to be located on conjugative plasmids belonging to different incompatibility groups [57,58,59]. Plasmids in *E. coli* can contain multiple replicons that are known to mediate resistance to antimicrobials and disinfectants as well as imparting increased virulence [60,61]. In this study, 64% of beef isolates harboured more than two plasmid replicons compared to pig isolates (82%), poultry (100%), and human (85%) isolates. The plasmid replicons IncFIB (AP001918) and IncFII were found to be strongly associated with multiple classes of ARGs. These included genes that confer resistance to ß-lactams (*bla*_TEM-1B_), aminoglycosides (*aph(3'')-Ib* and *aph(6)-Id*), tetracycline (*tet(A)* and *tet(B)*), trimethoprim (*dfrA5*), and sulfamethoxazole (*sul2* and *sul1*). The RepFIB/FIIA plasmids are known to contain and transfer multidrug resistance-encoding islands and VGs [62]. This highlights the potential of these plasmids to spread AMR across bacterial populations and contribute to the increasing threat of AMR. The plasmid IncFIB (AP001918) was also reported to be the most common replicon in *E. coli* isolated from pig, poultry, and human isolates [31,63,64]. The IncF plasmids (IB and II) are known to contain ARGs, such as the ß lactamase genes *bla*_TEM-1B_, *bla*_CMY-2_, and *bla*_OXA-1_, and those encoding resistance to sulfonamides and trimethoprim (*dfrA8*, *strA*, *strB*, and *sul2*), as well as the tetracycline resistance genes *tet(A)* and *tet(B)* [65,66,67,68]. Whilst previous studies have shown that plasmid replicons can be shared among *E. coli* isolated from humans and food animals [65], the present study showed that both the frequency and heterogeneity of ARGs and plasmids were much lower in beef cattle isolates compared to pig, poultry, and human isolates, thus confirming that feedlot beef cattle from this study population represent reduced AMR risk to public health and the environment within Australia.

The increased detection of *E. coli* that are resistant to extended-spectrum β-lactam antimicrobials has been linked to the dissemination of international high-risk clones, such as ST10, ST38, ST69, ST73, ST115, ST117, ST131, ST354, ST410, ST457, ST517, ST648, ST711, and ST1193, in animals and humans [69,70]. In this study, the three most common STs found in beef cattle isolates were ST10 (10/37; 27.0%), ST278 (7/37; 19.0%), and ST109 (2/37; 5.4%). As a cause of human blood stream infections, ST73 (9/40; 22.5%), ST95, and ST131 (7/40; 17.5, each) predominated, whereas among pig isolates ST361 (4/45; 8.9%) and ST48 and ST398 (3/45; 7.5% each) predominated, while among poultry isolates ST 117 (5/19; 26.3%) and ST57 and ST350 (2/19; 10.5%) were most frequently observed. The clonally diverse group ST10, which belongs to phylogenetic group A, is known to have a broad host range and can be found in a variety of hosts, including humans, animals, and the environment [14]. In the present study, the clonal complex ST10 was identified as the most common lineage among the beef cattle isolates, with a prevalence of 27.0% compared to 4.4% in pigs and 2.5% in humans. Similarly, ST10 was the most commonly identified isolate from commercial beef cattle farms in the United States and in isolates from the Australian beef cattle population at slaughter [44,71]. The CC10 *E. coli* has been reported to carry ARGs and can sometimes cause infections in dogs, humans, and pigs [39,72]. However, the transmission of AMR from food animals to humans (and vice versa) is still not fully understood. Some research has reported direct contact with animals as a potential source of transmission for zoonotic bacteria, such as ESBL-producing *E. coli*, while other studies suggest the transmission is more likely to occur from humans to animals [73,74,75]. However, the fact that only a single human clinical isolate in the present study belonged to ST10 indicates that in Australia it is not a significant opportunistic human pathogen regardless of host source.

This study had some limitations. Firstly, the study included seven ESBL-producing *E. coli* isolated from beef cattle using selective media. However, it is important to note that the use of selective media could have introduced a selection bias in amplifying these ESBL-producing *E. coli* isolates, which would not normally be selected as the most common colony type on less inhibitory agar. Secondly, the *E. coli* data from pigs, poultry, and humans used in the study were obtained from a universal database as secondary information; ideally, the studies should have been prospective in design and more synchronised. Thirdly, *E. coli* isolates obtained from pigs and poultry were selected on the basis of possessing Class 1 integrons, which is considered as an indication of having an antimicrobial resistance genotype, whereas most of the beef cattle isolates were selected on the basis of being resistant to one or more antimicrobials from a previous study [26]. Fourthly, the limited sample size, particularly the availability of only 19 poultry isolates that met the selection criteria for comparison, may not be as representative of larger populations; ideally more isolates would have provided firmer conclusions. Lastly, the short-read sequencing data analysed in this study did not allow for the identified ARGs and virulence genes to be mapped to specific plasmids, which requires long read sequencing [76,77].

## 4. Materials and Methods

### 4.1. Whole Genome Sequencing and Phylogenetic Analysis 

The current study was a follow-up to previous work focused on determining the prevalence of antimicrobial resistance among rectal *E. coli* isolates obtained at entry to and exit from an Australian beef cattle feedlot [28]. A total of thirty-three *E. coli* isolates determined to be resistant to at least one antimicrobial (including seven that were resistant to extended-spectrum β-lactams isolated from ESBL selective agar rather than MacConkey agar), together with four *E. coli* isolates that were susceptible to all tested agents, were randomly selected in the study. DNA extraction, whole genome sequencing (WGS), and genomic assembly have been described in our previously published research [28]. Raw genomic sequences were deposited in the NCBI under BioProject PRJNA844571. The whole-genome sequenced commensal *E. coli* isolated from beef cattle (n = 37; this study) were compared with retrieved genomic data from three previous studies undertaken in Australia. These included: (i) 45 *E. coli* isolates carrying the Class 1 integrase gene (*intI1*) isolated from sows and their offspring at a commercial pig breeding operation (accession number PRJNA509690) [78]; (ii) 19 avian pathogenic *E. coli* isolates that also carried *intI1*, which were originally obtained from post-mortem lesions from deceased or culled birds with signs of an APEC infection at different poultry operations in Australia (accession number PRJNA479542) [79]; and (iii) 40 *E. coli* isolates from bloodstream infections in humans in Australia (accession number PRJNA480723) [80]. Assembled sequences with less than 30x coverage and fewer than 25,000 SNPs were excluded from further analysis. 

Genetic relationships between isolates were examined using single nucleotide polymorphism (SNPs) found from cleaned WGS reads mapped to an *E. coli* complete genome (NCBI Assembly Accession: BA000007.3). The software Snippy v4.6.0 (https://github.com/tseemann/snippy (accessed on 11 September 2022)) was used to call core SNPs, i.e., SNPs that can be determined in all isolates. A maximum likelihood (ML) tree was constructed with RAxML v8.2.10 using the model GTRCAT and a rapid bootstrap analysis with 100 bootstraps for the best scoring ML tree [81]. This was followed by recombination removal using ClonalFrameML v1.12 [82]. The final phylogenetic tree and heat map were manipulated with iTOL (https://itol.embl.de/ (accessed on 11 September 2022)) for display [83]. A heat map illustrating the presence or absence of each trait for each isolate was created to assess all data elements for all isolates.

### 4.2. Determination of Genotypes, Antimicrobial Resistance Genes, and Virulence Genes

Multilocus sequence type (MLST) and phylogenetic group were determined using MLST 2.0 [84] and ClermonTyping [85], respectively. To identify ARGs, we used ResFinder 4.0 [86]. To further pinpoint the chromosomal point mutations associated with AMR, we used PointFinder [87]. The input to start search in ResFinder and PointFinder was the assembled *E. coli* genome. Additionally, AMR genes were predicted using the Antibiotic Resistance Genes Database (ARDB) and the Comprehensive Antibiotic Resistance Database (CARD) [88]. PlasmidFinder was used with a minimum identity of 95% and coverage of 60% to detect plasmid replicons, while pMLST 2.0 was employed to perform plasmid multi-locus sequence typing [89]. Virulence genes were identified using VirulenceFinder 2.0 [90,91]. IntegronFinder was utilized to identify integrons and their unique components (accessed on 28 March 2023) [92] and confirmed using BLAST with a minimum identity of 95% (https://blast.ncbi.nlm.nih.gov/Blast/ (accessed on 28 March 2023)) [93].

### 4.3. Statistical Analyses

Categorical measured traits including ARGs, disinfectant resistance genes, the presence of plasmid replicons, and VGs were converted into numerical code with 1 indicating presence and 0 indicating absence. The resistance profile was categorised as MDR if the isolate exhibited resistance to one or more antimicrobials in three or more antimicrobial classes [94]. AMR and disinfectant gene frequencies were described as rare: <0.1%; very low: 0.1% to 1.0%; low: >1% to 10.0%; moderate: >10.0% to 20.0%; high: >20.0% to 50.0%; very high: >50.0% to 70.0%; and extremely high: >70.0%, according to the European Food Safety Authority (EFSA) and the European Centre for Disease Prevention and Control (ECDC) [95].

## 5. Conclusions

This study highlighted the relative abundance of ARGs (to both antimicrobials and disinfectants), plasmid replicons, and virulence-associated genes in commensal *E. coli* isolated from Australian beef cattle in comparison to additional commensals and extraintestinal pathogens isolated from food-producing animals (pigs and poultry, respectively) and extraintestinal pathogens from humans. The results confirmed that beef cattle represent a reduced risk to public health and the environment given that most isolates belonged to phylogenetic groups typically associated with commensal bacteria. Whilst a significant proportion of beef cattle isolates were ESBL/*AmpC* producing *E. coli* (some of which belonged to broad host range ST10), there was limited evidence that these higher risk clones are problematic in human medicine in Australia. 

## Figures and Tables

**Figure 1 antibiotics-12-00895-f001:**
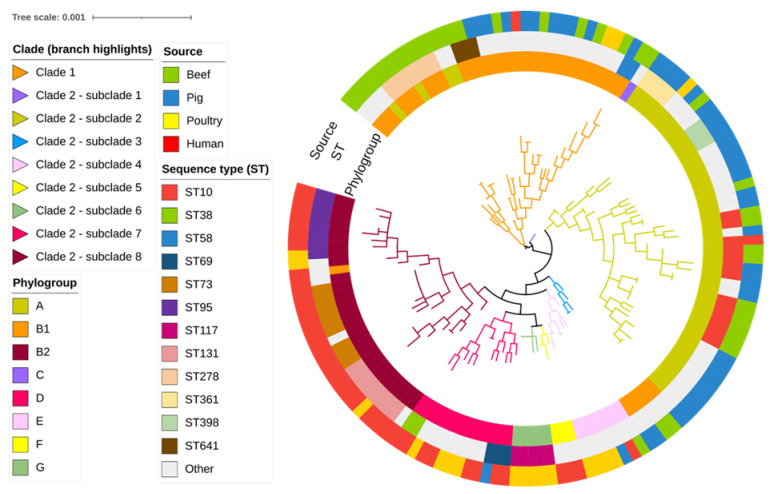
A mid-point rooted, maximum-likelihood phylogenetic tree constructed based on analysis of single-nucleotide polymorphisms (SNPs) of the core SNPs of 141 *E. coli* genomes isolated from beef (n = 37), pig (n = 45), poultry (n = 19), and human (n = 40) sources. Branches are coloured by clade and subclade according to the legend. Phylogroups (inner ring), sequence types (STs) (middle ring), and sources of the isolates (outer ring) are annotated according to the legend.

**Figure 2 antibiotics-12-00895-f002:**
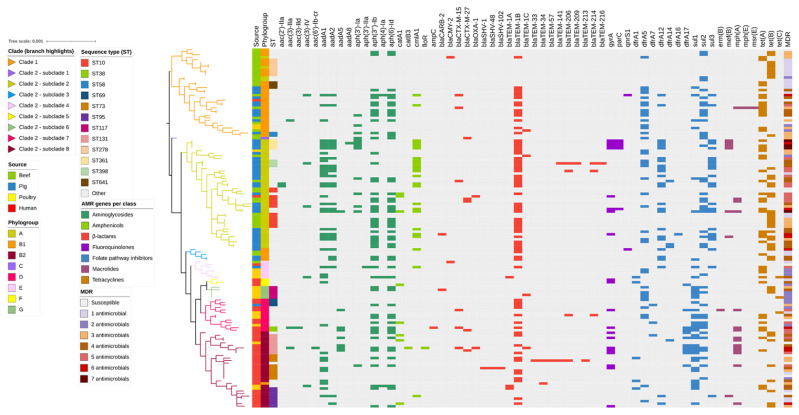
Antimicrobial resistance gene (ARG) profiles clustered on the basis of the Figure 1 SNP-based phylogenetic tree composed of 141 *Escherichia coli* genomes isolated from beef cattle (n = 37), pig (n = 45), poultry (n = 19), and human (n = 40) sources. Branches are coloured by clade and subclade, according to the legend. The remaining columns indicate: (1) isolate source, (2) isolate phylogroup, (3) isolate sequence type, (4) ARGs detected in the isolate, and (5) multidrug resistance profile of the isolate. The detected ARGs are clustered according to their respective antimicrobial classes, as follows: aminoglycosides (14 ARGs shown in dark green), amphenicols (4 ARGs shown in light green), β-lactams (21 ARGs shown in red), fluoroquinolones (3 ARGs shown in purple), folate synthesis inhibitors (10 ARGs shown in blue), macrolides (5 ARGs shown in magenta), and tetracyclines (3 ARGs shown in brown).

**Figure 3 antibiotics-12-00895-f003:**
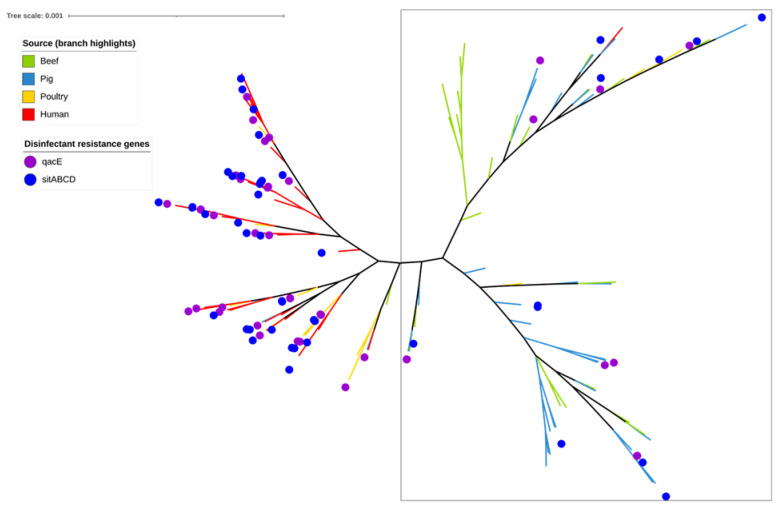
SNP-based phylogeny of 141 *Escherichia coli* isolates isolated from beef cattle (green branches, n = 37), pigs (blue branches, n = 45), poultry (orange branches, n = 19), and human sources (red branches, n = 40). Purple circles indicate the isolates shown to contain the *qacE* disinfectant resistance gene (imparting resistance to quaternary ammonium compounds), whereas the blue circles indicate that the isolates shown contain the *sitABCD* gene (imparting resistance to hydrogen peroxide). Most beef and pig isolates are located within phylogroups A and B1, which are encompassed within the square box.

**Figure 4 antibiotics-12-00895-f004:**
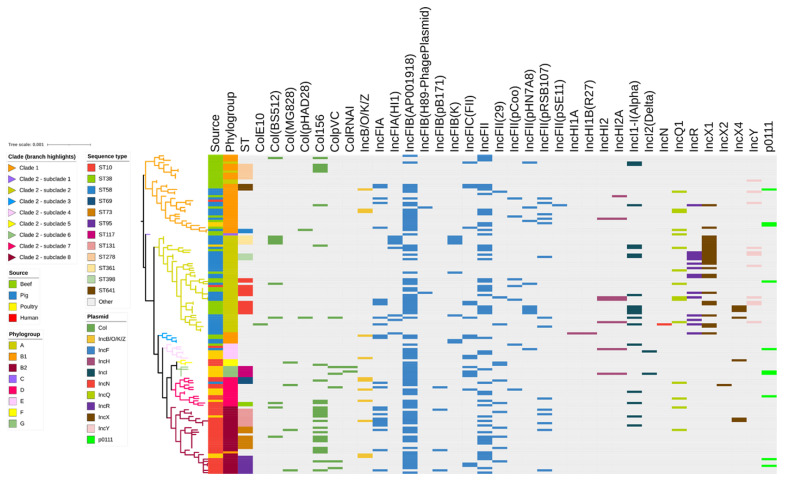
Plasmid replicon profiles clustered on the basis of the Figure 1 SNP-based phylogenetic tree composed of 141 *Escherichia coli* genomes isolated from beef cattle (n = 37), pig (n = 45), poultry (n = 19), and human (n = 40) sources. Branches are coloured by clade and subclade according to the legend. The remaining columns indicate: (1) isolate source, (2) isolate phylogroup, (3) isolate sequence type, and (4) plasmid replicon detected in the isolate. The detected plasmids are clustered according to their respective types, as follows: Col (7 plasmid replicons shown in dark green), IncB/O/K/Z (1 plasmid replicon shown in yellow), IncF (13 plasmid replicons shown in blue), IncH (4 plasmid replicons shown in light purple), IncL (2 plasmid replicons shown in dark blue), IncN (1 plasmid replicon shown in dark red), IncQ (1 plasmid replicon shown in light green), IncR (1 plasmid replicon shown in purple), IncX (3 plasmid replicons shown in dark brown), IncY (1 plasmid replicon shown in light red), and p0111 (1 plasmid replicon shown in green).

**Figure 5 antibiotics-12-00895-f005:**
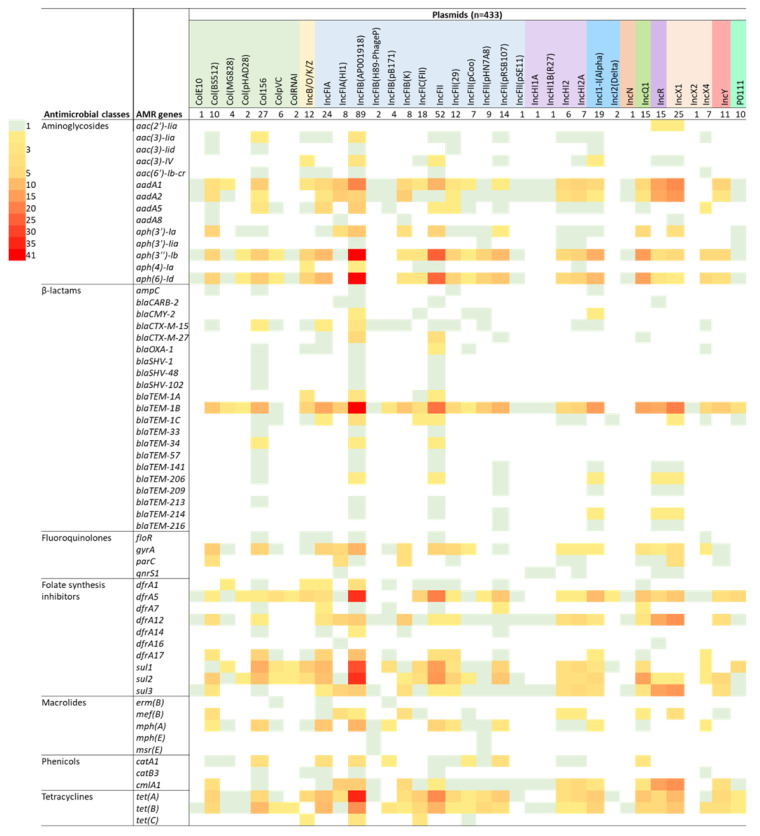
Cross-tabulation heat map showing the degree of correlation between plasmid replicon types and the presence of an antimicrobial resistance gene (ARG) for 141 isolates of *Escherichia coli* isolated from beef cattle (n = 37), pig (n = 45), poultry (n = 19), and human (n = 40) sources. Plasmid replicons are listed horizontally (total number identified in the 141 isolates is also indicated), whereas the ARGs are listed vertically in their classes. The colour strips indicate the number of isolates (in multiples of 5) exhibiting each particular plasmid/ARG match.

**Figure 6 antibiotics-12-00895-f006:**
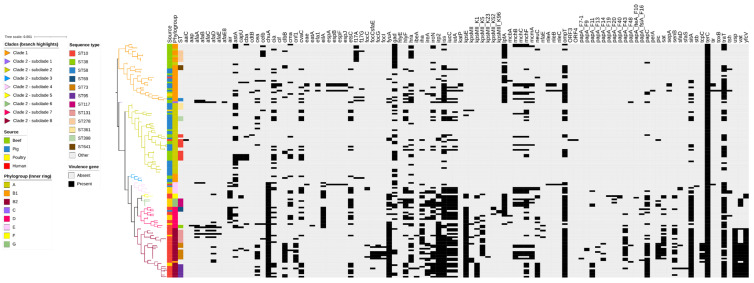
Virulence gene (VG) profiles clustered on the basis of the Figure 1 SNP-based phylogenetic tree composed of 141 *Escherichia coli* genomes isolated from beef cattle (n = 37), pig (n = 45), poultry (n = 19), and human (n = 40) sources. Branches are coloured by clade and subclade according to the legend. The remaining columns indicate: (1) isolate source, (2) isolate phylogroup, and (3) isolate sequence type. The remaining columns refer to the presence (black) or absence (grey) of 103 specific VGs identified in the collection.

**Table 1 antibiotics-12-00895-t001:** The frequency of different phylogroups in *E. coli* isolated from beef cattle, pig, poultry, and human sources.

Sample Source	Phylogroup (%)							
	A	B1	B2	C	D	E	F	G
Beef (n = 37)	16 (43.2)	19 (51.3)	0	1 (2.7)	0	1 (2.7)	0	0
Pig (n = 45)	29 (64.4)	14 (31.1)	0	0	1 (2.2)	1 (2.2)	0	0
Poultry (n = 19)	1 (5.3)	2 (10.5)	3 (15.8)	0	4 (21.0)	4 (21.0)	0	5 (26.3)
Human (n = 40)	1 (2.5)	2 (5.0)	26 (65.0)	0	8 (20.0)	0	3 (7.5)	0

**Table 2 antibiotics-12-00895-t002:** The diversity of ß-lactam resistance genes identified among 141 *E. coli* isolated from beef cattle, pig, poultry, and human sources.

β-Lactam	Sample Source							
Resistance Genes	Beef Cattle (n = 37)		Pig (n = 45)		Poultry (n = 19)		Human (n = 40)	
	Frequency (%)	Phylogroup (n)	Frequency (%)	Phylogroup (n)	Frequency (%)	Phylogroup (n)	Frequency (%)	Phylogroup (n)
*ampC*	-	-	-	-	-	-	1 (2.5)	D (1)
*bla* _CARB-2_	-	-	1 (2.2)	A (1)	1 (5.3)	G (1)	-	-
*bla* _CMY-2_	2 (5.4)	A (1), E (1)	-	-	-	-	-	-
*bla* _CTX-M-15_	3 (8.1)	A (1), B1 (2)	-	-	-	-	3 (7.5)	B2 (1), D (2)
*bla* _CTX-M-27_	3 (8.1)	A (2), C (1)	-	-	-	-	2 (5.0)	B2 (1), D (1)
*bla* _OXA-1_	-	-	-	-	-	-	2 (5.0)	A (1), B2 (1)
*bla* _SHV-1_	-	-	-	-	-	-	1 (2.5)	B2 (1)
*bla* _SHV-48_	-	-	-	-	-	-	1 (2.5)	B2 (1)
*bla* _SHV-102_	-	-	-	-	-	-	1 (2.5)	B2 (1)
*bla* _TEM-1A_	-	-	-	-	3 (15.8)	B2 (2), E (1)	-	-
*bla* _TEM-1B_	8 (21.6)	A (6), B1 (2)	42 (93.3)	A (26), B1 (14), D (1), E (1)	3 (15.8)	A (1), B1 (1), D(1)	17 (42.5)	B2 (12), D (5)
*bla* _TEM-1C_	2 (5.4)	A (1), B1 (1)	-	-	1 (5.3)	G (1)	1 (2.5)	B2 (1)
*bla* _TEM-33_	-	-	-	-	-	-	1 (2.5)	B2 (1)
*bla* _TEM-34_	-	-	-	-	-	-	2 (5.0)	B1 (1), B2 (1)
*bla* _TEM-57_	-	-	-	-	-	-	1 (2.5)	B2 (1)
*bla* _TEM-141_	-	-	1 (2.2)	A (1)	-	-	1 (2.5)	B2 (1)
*bla* _TEM-206_	-	-	2 (4.4)	A (2)	1 (5.3)	D (1)	1 (2.5)	B2 (1)
*bla* _TEM-209_	-	-	1 (2.2)	A (1)	-	-	-	-
*bla* _TEM-213_	-	-	-	-	-	-	1 (2.5)	B2 (1)
*bla* _TEM-214_	-	-	2 (4.4)	A (2)	1 (5.3)	D (1)	-	-
*bla* _TEM-216_	-	-	1 (2.2)	A (1)	-	-	-	-

## Data Availability

WGS reads are available in the SRA under BioProject; beef (PRJNA844571), pig (PRJNA509690), poultry (PRJNA479542), and human (PRJNA480723) isolates.

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
