# Peer review of "Phylogenetic Analysis of Escherichia coli Isolated from Australian Feedlot Cattle in Comparison to Pig Faecal and Poultry/Human Extraintestinal Isolates"

_antibiotics, 2023, doi:10.3390/antibiotics12050895_

Round 1

Reviewer 1 Report

The authors describe genetic characteristics and phylogenetic relationships among E. coli from beef cattle, pigs, poultry and humans based on whole genome sequencing data. The findings indicate that E. coli from beef cattle represent a reduced risk to public health. Additionally, there is limited evidence that higher risk clones from cattle are problematic in human medicine. These conclusions are highly relevant for experts in the field. The study seems to be scientifically sound, and the results are presented in a clear and concise manner. However, there are a few points that need to be addressed (at least more intensively).

Line 127: It is very amazing that poultry isolates have lower MDR rates than pig isolates. This result must be critically discussed, based on literature. There are many factors that might have influenced this result which is certainly not generally valid. For example, poultry (pathogenic) E. coli were isolated from post-mortem lesions, whereas the other isolates origin from feces (commensal; pigs, beef cattle) and from blood (pathogenic; humans). Pathogenic isolates might lose their resistance genes in favor of the virulence genes (“shedding ballast”). It is also known that there are big differences between laying hens/broiler chickens, conventional/organic husbandry systems etc. considering antibiotic resistance. Additionally, a sample number of n = 19 is relatively low. Admittedly, all points are discussed in the manuscript, but in this special context, it should be discussed more intensively and not scattered throughout the text. In any case, the reduced informative value of this result should be pointed out.

The entire manuscript should be carefully checked for missing (primary) lecture. A few examples:

Line 39: who states that many E. coli sub-types are relatively host-specific?

Line 237: Authors should cite references dealing with E. coli, not with Enterococci (e.g., doi: 10.1016/j.vetmic.2009.09.066.).

Line 258: The findings of the authors are in line with some other (European) studies, but there are also some slightly different results (e.g., doi: 10.1016/j.vetmic.2012.06.010.).

Line 358: provide reference or statistical basis for the statement that “twice as many isolates would have provided firmer conclusions”. Isn’t twice as many still a relatively low number?

Line 359: Provide references for this statement.

Minor points:

E. coli in italics throughout the text

Line 319: delete one “the”.

Reviewer 2 Report

The manuscript of Messele et al presents a comparative study of genetic similarity, ARG, plasmid, and VG profile of cattle-origin E. coli isolates to isolates from other food animals and humans in Australia.

The introduction content is succinctly described and contextualized with respect to previous and present theoretical background on the topic and supported by relevant references on the topic. Objectives of the study are clearly defined.

The methodology used is appropriate and adjusted to the objectives of this work and presented with detail. Results are presented clearly and properly analyzed

The article is original and have practical interest regarding antimicrobial-resistant E. coli of clinical importance. This study has some limitations (also mentioned by the authors), I recommend the acceptance of the manuscript after minor revision namely:

-       Revision of the whole document concerning the writing of Escherichia coli or E. coli in italic, along the document several time the italic is missing.

-       Regarding the authors affiliation 4 is repeated and 5 is missing  .

Reviewer 3 Report

Dear authors,

In this manuscript you show similarities between commensal strains of E. coli isolated from healthy cattle and extraintestinal strains of E. coli isolated from pigs, poultry and humans, by using modern molecular methods. Also, you reported the findings of mentioned strains in relation to phylogenetic groups, sequence types, antimicrobial and disinfectant resistance genes, plasmid replicons and virulence genes.

The paper is very well written, it is possible to follow the Materials and Methods and the obtained results, and the Discussion connects the findings of previous and this research in  Australia, but also more widely when it was possible.

Specific comments: is there any reason why E. coli was not written in Italic in the chapter Results? (in all other chapters it is written in Italic), so please, uniform the writing in the whole manuscript.
